# Variability of Coordination in Typically Developing Children Versus Children with Autism Spectrum Disorder with and without Rhythmic Signal

**DOI:** 10.3390/s20102769

**Published:** 2020-05-13

**Authors:** Lidia V. Gabis, Shahar Shefer, Sigal Portnoy

**Affiliations:** 1Department of Occupational Therapy, Sackler Faculty of Medicine, Tel Aviv University, Tel Aviv 6997801, Israel; Lidia.Gabis@sheba.health.gov.il; 2The Weinberg Child Development Center, Safra Children’s Hospital, Sheba Medical Center, Ramat Gan 52621, Israel; drshahar.shefer@sheba.health.gov.il

**Keywords:** motor variability, actigraphy, triaxial accelerometers, jumping

## Abstract

Motor coordination deficit is a cardinal feature of autism spectrum disorder (ASD). The evaluation of coordination of children with ASD is either lengthy, subjective (via observational analysis), or requires cumbersome post analysis. We therefore aimed to use tri-axial accelerometers to compare inter-limb coordination measures between typically developed (TD) children and children ASD, while jumping with and without a rhythmic signal. Children aged 5–6 years were recruited to the ASD group (n = 9) and the TD group (n = 19). Four sensors were strapped to their ankles and wrist and they performed at least eight consecutive jumping jacks twice: at a self-selected rhythm and with a metronome. The primary outcome measures were the timing lag (TL), the timing difference of the maximal acceleration of the left and right limbs, and the lag variability (LV), the variation of TL across the 5 jumps. The LV of the legs of children with ASD was higher compared to the LV of the legs of TD children during self-selected rhythm jumping (*p* < 0.01). Additionally, the LV of the arms of children with ASD, jumping with the rhythmic signal, was higher compared to that of the TD children (*p* < 0.05). There were no between-group differences in the TL parameter. Our preliminary findings suggest that the simple protocol presented in this study might allow an objective and accurate quantification of the intra-subject variability of children with ASD via actigraphy.

## 1. Introduction

Autism spectrum disorder (ASD) is defined by impairments in social communication and/or the presence of repetitive or restricted behaviors. The prevalence of ASD in the United States is reported as one in 68 children [1]. Aside from social difficulties, motor impairments are prevalent in individuals with ASD and worsen with age [2]. Motor impairments in individuals with ASD might affect both gross and fine motor functions, e.g., manual dexterity and balance [3]. Specifically, motor coordination deficit was characterized as a cardinal feature of ASD [3].

To date, the accuracy level of the diagnostic tests of ASD are relatively limited [4]. Specifically, standardized motor assessments for children with ASD take between 15 min to over one hour to complete [5]. It has been recently suggested that simple quantitative measures of motor coordination may assist in the identification of subtle motor impairments in individuals with ASD [5,6]. Early detection of abnormalities in the coordination abilities of the child may assist clinicians in devising an optimal treatment plan, e.g., engaging children with ASD in ball games [7]. Therefore, devising a quick and simple protocol for an examination that produces accurate quantitative measures of the child’s coordination capabilities is a challenge for future studies.

Previous studies characterizing coordination abnormalities in children have attempted to use rhythmic signals, e.g., via a metronome, to analyze movement synchronization. For example, coordination abnormalities in children with ASD were measured while performing various multi-limb actions with a metronome, such as marching and clapping [8]. The coefficient of variation (CV) of the inter-event duration was obtained as a variability measure by analyzing recorded video data. The authors reported that children with ASD exhibit higher CV compared to typically developed (TD) children, but there was no difference in the CV between two groups of children with ASD, with lower and higher intelligence quotient (IQ) [8]. However, using video recordings of the child in the clinical setting might not be appreciated by their parents. Moreover, the post analysis is cumbersome and does not produce quick results.

Wearable tri-axial accelerometry is a simple and effective mean to record activity in individuals with ASD. Actigraphy was previously used to report physical activity levels in individuals with ASD [9] and the effects of various factors (for example age [10], social engagement with adults [11], and household structure [12]) on the physical activity levels of these individuals. Additionally, actigraphy has been used to monitor sleep patterns in individuals with ASD [13,14], demonstrating, for example, that sleep latency, as measured by actigraphy, was longer in individuals with ASD compared to controls [15]. Another study showed that an accelerometer worn by youth with ASD can predict aggression to others, one minute before it occurs [16]. Overall, the literature supports the usage of wearable tri-axial accelerometers with individuals with ASD. However, to the best of our knowledge, no studies utilized these sensors to assess lower and upper limb coordination in children with ASD while performing a quick and simple jumping activity. Furthermore, rhythmic auditory cueing was suggested as a technique to stabilize the variability in the movement pattern and facilitate a motor plan for individuals with ASD [17]. Since the effect of a rhythmic signal on the coordination measures while performing jumping activity has not been reported, our aim was to compare inter-limb coordination measures between TD children and children ASD, while jumping with and without a rhythmic signal.

## 2. Methods

### 2.1. Population

We recruited children, aged 5–6 years, diagnosed with ASD according to the criteria listed in the Diagnostic and Statistical Manual of Mental Disorders, Fifth Edition (DSM-V). Exclusion criteria were inability to understand and/or comply with simple instructions, other pathologies, e.g., epilepsy, orthopedic impairments, uncorrected auditory or visual impairments, cognitive disability. Age-matched TD children were recruited as controls. The study received the approval of the Helsinki committee of the hospital (approval #0119-13-SMC).

### 2.2. Tools

Four Actigraph™ sensors (GT3X; ActiGraph, Pensacola, FL, USA) were used in this study. These are small (3.8 × 3.7 × 1.8 cm) and lightweight (27 g) sensors, easily donned on the limbs of the subjects using elastic belts. The sensors provide accelerations in three axes and can be activated at a frequency of 30 Hz. These sensors have been extensively used in various populations of different ages, as recently reviewed in [18]. Specifically, the sensors have been used in children with ASD [12]. For the rhythmic signal, a digital metronome was set to 1 Hz. This frequency was chosen following a pilot with TD children, by setting the metronome to various frequencies and asking the children which was their preferred comfortable choice of jumping frequency.

### 2.3. Protocol

The parents signed an informed consent form and the child gave verbal consent to participate in the trial. The parents filled out a demographic questionnaire. Then, two sensors were attached to the wrists of the child and two to the ankles using elastic belts. The child was asked to ambulate with the sensors and report any inconvenience. Then, the child was asked to perform at least 8 consecutive jumping jacks (also called star jumps). The jumps were demonstrated before data recording began. This was performed twice: once at a self-selected rhythm and once guided by the metronome, located 1 m behind the subject.

### 2.4. Post Analysis

The data from each accelerometer were downloaded to a personal computer and then exported as CSV files using Actilife™ software version 5.10.0 (ActiGraph, Pensacola, FL, USA). A custom code was created in LabView (v2015, National Instruments, Austin, TX, USA). The acceleration magnitude was calculated using the accelerations of the 3 axes, *A_x_*, *A_y_*, and *A_z_*, as:(1)$$Limb\;acceleration\;\left[\frac{m}{s^{2}}\right]=\sqrt{A_{x}^{2}+A_{y}^{2}+A_{z}^{2}}$$

Five consecutive jumps were taken (the first and last jumps were excluded from the analysis as initiation and termination of movement). We designed this protocol to include a small number of jumps because a longer jumping sequence might involve fatigue, which will produce bias in the results of limb accelerations. Also, the cooperation levels and concentration span of 5-year-olds in the clinical settings might be low, so that a longer examination might not be possible. The maximal acceleration of each limb of each jump was calculated. Also, the timing of the peak acceleration (in seconds) of each jump was calculated and the following inter-limb timing measures were computed using the following formulas:(2)$$TL\;\left[sec\right]=\frac{\sum \left(t_{L}-t_{R}\right)}{5}$$
(3)$$LV\;\left[sec\right]=\sqrt{\frac{\sum TL^{2}-\frac{{\left(\sum TL\right)}^{2}}{5}}{5}}$$

The timing lag (*TL*) is the difference in the timing of the maximal acceleration of the left lower or upper limb ($t_{L}$) and right lower or upper limb ($t_{R}$), averaged for five consecutive jumps. It is defined similarly as constant error of two measures (left and right limb herein) that are expected to be identical during symmetric movement [19,20]. A positive TL value denotes that the left limb reached maximal acceleration sooner compared to the right limb. As the TL value decreases towards zero, the coordination of the two limbs in reaching maximal acceleration is higher, meaning that they are more in-phase. The lag variability (*LV*) is the variation of TL across the 5 jumps. It is defined in a similar manner as variable error of two measures and produces the average of the standard deviation [19]. High LV is indicative of low consistency between jumps. These measures are calculated separately for each condition, with and without the metronome rhythm, and presented in seconds.

All of the statistical analyses were performed in IBM SPSS Statistics 25. Mann–Whitney U test was used to test for between-group differences in age and body mass index (BMI), and the Chi-square test was used to test for between-group differences in sex. ANOVA analysis of group (ASD and TD) × condition (with and without metronome) was performed. Post hoc tests were administered according to the findings. Effect size estimates, *r*, for Mann–Whitney non-parametric tests were calculated according to [21]:(4)$$r=\frac{Z}{\sqrt{N}}$$

Statistical significance was set to *p* < 0.05.

## 3. Results

The personal characteristics of the two groups are detailed in Table 1. There were no significant differences between the two groups in age, sex, and BMI.

The maximal limb accelerations of each group in each condition, as well as the coordination parameters of the upper and lower limbs are presented in Table 2. There was a significant main effect of group, but there was no main effect of condition. Specifically, the TD children reached higher accelerations of their left limbs during self-selected rhythm jumping compared to children with ASD (Figure 1). In that condition, the LV of their legs were lower compared with the LV of the legs of children with ASD (Figure 2). Additionally, the LV of the arms in TD children, jumping with the rhythmic signal, was lower compared to that of the ASD group (Figure 2).

## 4. Discussion

We compared inter-limb coordination measures between TD children and children with ASD, while performing jumping jacks with and without rhythmic signal. Our main findings show no effect of the rhythmic signal on the coordination measures. However, children with ASD exhibited high variability in limb coordination, as shown by the LV measure, compared to the TD children. This is the first study to report a difference in an inter-limb coordination measure between children with ASD and TD children, performing a simple, quick four-limb jumping activity.

While the maximal value of the limb accelerations was not a primary outcome measure of this study, we report lower accelerations in the ASD group compared to the TD group (statistically significant for the left leg and arm during self-selected rhythm jumps and not significant but showing the same trend for the other limbs and the rhythmic signal condition). This finding is expected, as several publications report slower repetitive hand and foot movement assessed with standardized test batteries in individuals with ASD, as reviewed by Gowen and Hamilton [22]. Also, drumming movements of children with ASD were reported as slower compare to TD children [23].

There was no statistically significant main effect of the rhythmic signal, provided during the jumping activity. Since there are reports of impaired early auditory pathways in ASD [24], the timing of motor neuron transmission in our ASD group may have been influenced by delayed auditory processing, rendering the cues unhelpful or even disturbing to the task execution. This explanation was suggested in a study that compared the cadence during an auditory-cued two-legged hopping task between TD and ASD groups [25]. While the TD group showed a high performance of synchronizing their jumps with the cues, the ASD group showed a varied deviant response to the cueing [25]. In our study, however, we did not test for synchronization between the rhythmic signal and the movements of the children. Therefore, we cannot attest to the success or failure of the metronome in regulating the jumping sequence. Our results suggest that the rhythmic signal has no effect on the inter-limb coordination or its variability between jumps in children (with or without ASD) while performing jumping jacks.

Surprisingly, we found no statistically significant differences in the TL outcome measure between the two groups. This finding suggests similar inter-limb coordination between TD and ASD children performing jumping jacks. We assume that this finding can be explained by the simplicity of the chosen activity. Contrarily to marching or drumming activities, which involve out-of-phase inter-limb movement, jumping jacks comprise of in-phase symmetrical limb movements. It was suggested that this type of activity produces a simultaneous activation of homologous muscle groups [26]. Therefore, the complexity of motor planning required for the activity chosen for this study might be smaller compared to activities such as gait. We assume that the deficits in motor planning in children with ASD contributed to the inter-limb coordination deficits in the studies reported in the literature due to the more complex task chosen. Conversely, for the jumping jacks activity, we surmise that the main factor influencing coordination is not motor planning, but the sense of proprioception. The ability to perform inter-limb coordinated movements relies, among other factors, on an intact sense of proprioception [27]. It has been reported that the sensory input of individuals with ASD is intact. Specifically, studies demonstrated no deficit in proprioception in individuals with ASD. For example, the accuracy and precision of the proprioceptive estimates of identifying the angle of the elbow and the position of the fingertip in adolescents with ASD was similar to adolescents without ASD [28]. We therefore conclude that the similarity in the TL between the ASD and TD groups relates to the characteristics of the jumping activity, selected for this study, which relies more on the sense of proprioception then on motor planning.

As expected, children with ASD exhibited high variability in limb coordination, i.e., high LV measures, compared to the TD children. This means that while the inter-limb asynchronization in TD children was consistent across consecutive jumps, the inter-limb asynchronization in children with ASD varied between consecutive jumps, and this difference was statistically significant. High intra-individual variability is considered a marker for ASD [29,30]. The high intra-individual variability in ASD was demonstrated for measures such as reaction time [31,32], hand grip strength [29], finger tapping [29], drumming [23], and walking tasks [29,33,34]. Our research makes an important contribution to the literature as it demonstrates the ability to perform objective quantitative discrimination between TD children and children with ASD using a simple and quick protocol of a four-limb activity. This protocol could be used in future studies to investigate differences in intra-subject variability between groups of children with ASD of different sex, age, and level of IQ.

The main limitation of this study is the small sample size of the ASD group. Future studies should be encouraged by our preliminary results and continue the investigation on a larger population of children with ASD. Another limitation concerns the placement of the sensors, attached to the wrist and ankles of the subject. Although there were no between-group differences in BMI, slight variability of limb length between the children is expected. Accordingly, the maximal acceleration values may have been influenced by this, so that higher acceleration values would be measured when the sensor is located further from the shoulder or hip joint. This limitation, however, has no effect on the values of the TL and the LV since the calculations of these measures consider the difference between both limbs. Also, the metronome frequency, set to 1 Hz, might not have been suited to all participants and they might have ignored it. Finally, although we report coordination deficits in children with ASD, these could be attributed to differences in IQ, motivation, or imitation ability.

In conclusion, our preliminary findings suggest that the simple protocol presented in this study might allow an objective and accurate quantification of the intra-subject variability of children with ASD via actigraphy. This method should be further explored to discern between groups of children with ASD and other populations with motor dysfunction.

## Figures and Tables

**Figure 1 sensors-20-02769-f001:**
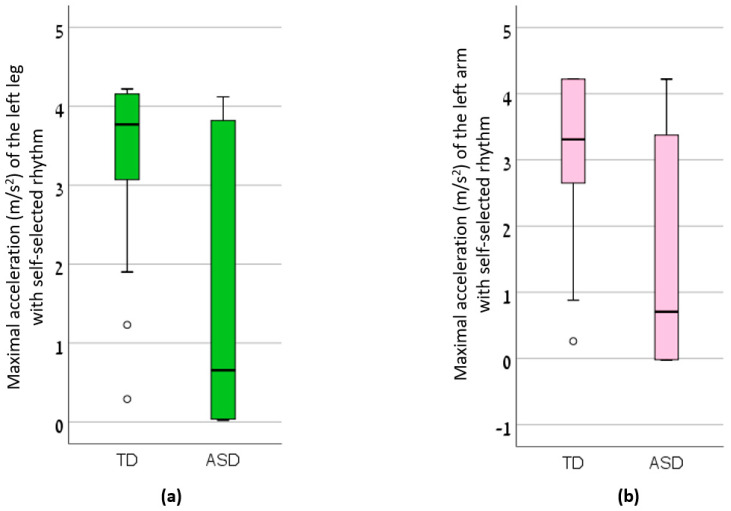
The maximal acceleration (m/s^2^) of the (**a**) left leg with self-selected rhythm and (**b**) left arm with self-selected rhythm in Typically Developed (TD) children and children with Autism Spectrum Disorder (ASD).

**Figure 2 sensors-20-02769-f002:**
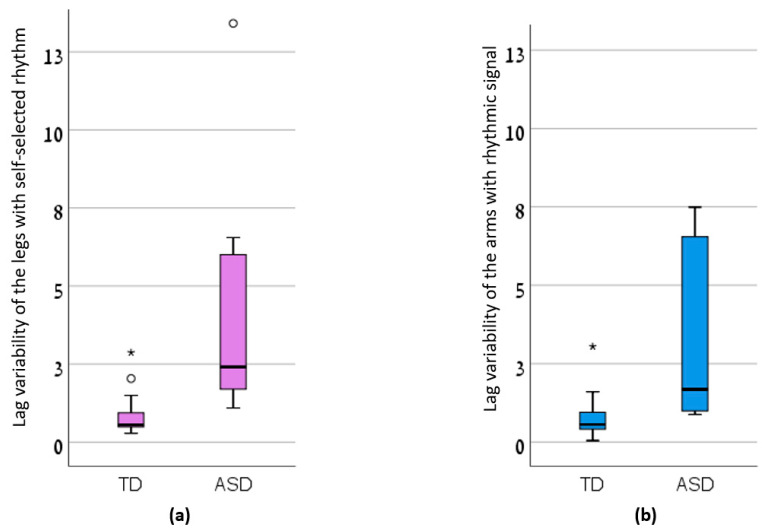
The lag variability of the (**a**) legs with self-selected rhythm and (**b**) arms with rhythmic signal in Typically Developed (TD) children and children with Autism Spectrum Disorder (ASD).

**Table 1 sensors-20-02769-t001:** Median and interquartile ranges of demographic characteristics of the subjects.

	TD (n = 19)	ASD (n = 8)	*p* Value
Age (years)	5.0 (4.4–6.0)	5.2 (5.0–6.6)	0.322
Sex	8 girls, 11 boys	2 girls, 6 boys	0.395
BMI (kg/m^2^)	14.4 (13.6–16.7)	15.4 (13.4–16.7)	0.710

TD: Typically Developed; ASD: Autism Spectrum Disorder; BMI: Body Mass Index.

**Table 2 sensors-20-02769-t002:** Medial and interquartile percentage of the maximal acceleration values (m/s^2^) of each limb and the coordination measures for the time of the peak acceleration (sec) of the arms and legs of each group, with and without the rhythmic signal.

	TD (n = 19)		ASD (n = 8)		Between-Groups F,p	Effect Size r
	Self-Selected Rhythm	Rhythmic Signal	Self-Selected Rhythm	Rhythmic Signal		
Left arm Acc.	**3.31 (2.49–4.22) ***	2.38 (1.35–4.13)	**0.71 (0.02–3.55) ***	0.68 (0.02–3.26)	6.444, 0.018	−0.412
Left leg Acc.	**3.77 (3.06–4.17) ***	3.78 (1.60–4.22)	**0.66 (0.03–3.94) ***	0.69 (0.10–3.51)	6.324, 0.019	−0.445
Right arm Acc.	3.26 (1.45–3.91)	1.93 (0.80–3.69)	1.89 (0.07–3.50)	0.91 (0.09–3.14)	NS	NS
Right leg Acc.	3.51 (3.28–3.96)	3.44 (2.10–4.14)	0.73 (0.02–4.06)	0.54 (0.01–3.78)	NS	NS
TL arms	0.13 (−0.78–0.64)	0.05 (−0.49–1.11)	0.87 (−0.35–2.86)	1.22 (−1.68–2.26)	NS	NS
TL legs	0.22 (−0.87–0.57)	0.61 (−0.08–0.91)	1.30 (−3.00–2.73)	0.64 (−4.28–3.17)	NS	NS
LV arms	0.61 (0.39–1.31)	**0.56 (0.41–0.98) ***	1.53 (0.88–3.15)	**1.68 (0.94–6.55) ***	9.812, 0.004	−0.583
LV legs	**0.55 (0.46–0.97) ****	0.58 (0.34–0.95)	**2.41 (1.68–6.28) ****	1.80 (0.55–3.20)	18.707, <0.001	−0.685

TD: Typically Developed; ASD: Autism Spectrum Disorder; Acc: Acceleration; TL: Timing Lag (see formula 2); LV: Lag Variability (see formula 3); NS: No Significance. Between group post hoc Mann-Whitney results in bold (* *p* < 0.05 and ** *p* < 0.01).

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
