# Peer review of "Variability of Coordination in Typically Developing Children Versus Children with Autism Spectrum Disorder with and without Rhythmic Signal"

_sensors, 2020, doi:10.3390/s20102769_

Round 1

Reviewer 1 Report

The paper presents a simple experiment involving children with ASD. The experiment was aimed at checking movement synchronization and coordination abnormalities when jumping with and without rhythmic signal. The acceleration of the movement was investigated, and based on it and two measures, some conclusions have been drawn.

There were eight children with ASD and 19 typically developed ones recruited for the experiment. The authors, during the analysis of the obtained results, found some significant differences between groups. in conclusion they stated: "the simple protocol presented in this study allows an objective and accurate quantification of the intra-subject variability of children with ASD via actigraphy". In my opinion, this is a too strong statement. Also, too few participants were engaged to state this, and too few jumps were analyzed.

Another question is if there are other quantities that can be extracted from registered signals to assess movement coordination. The meaning of LV should also be explained more precisely. Why was it defined in the presented way ?
It is not clear if the last column, in table 2, does show results of the comparison between subjects or between groups?

Reviewer 2 Report

The overall experiment design looks good and the obtained results are interesting and easy to reproduce, so I suggest to publish the article after minor revision. However, I would suggest to add some graphical information to the article: with the scarce sampling the statistics will be much more clear if it will be shown in the graphical form, not only as table of average quantities and its variations. I suggest to show raw data points either in small supplementary, or in the article directly and also show the results from the main table also as separate graphs. There is also probably typo "make" -> "male" in the Discussion section of the article.

Round 2

Reviewer 1 Report

Dear Authors,

Thank you for your response to my comments. Now, I think the manuscript explains my doubts.